# Comparative study of Taqman-based qPCR assay for the detection of *Anisakis simplex* and *Pseudoterranova decipiens*

**Mi-Gyeong Kim**[1], **Min Ji Hong**[1], **Doo Won Seo**[1*], **Hyun Mi Jung**[2], **Hyun-Ja Han**[1], **Seung Hwan Kim**[1], **Insun Joo**[1]

**1** Food Microbiology Division, National Institute of Food and Drug Safety Evaluation, Ministry of Food and Drug Safety, Cheongju-si, Chungcheongbuk-do, Republic of Korea, **2** National Institute of Fisheries Science, Incheon, Republic of Korea

* realize17@korea.kr

## Abstract

Anisakidosis is a foodborne parasitic infection caused by the consumption of raw or uncooked seafood that contains third stage larvae from the Anisakidae family. This infection has been observed across the globe, with a particularly high prevalence in South Korea and Japan. Consequently, there is a necessity to compare and analyze the optimal detection methods with a view to preventing Anisakis outbreaks. In this study, a species-specific Taqman-based qPCR method was developed for the detection of the internal transcribed spacer region and mtDNA genes of *Anisakis simplex* and *Pseudoterranova decipiens*. Parasite-specific primer/probe sets were selected based on the data from domestic and foreign detection methods. In addition, we have designed our own primer/probe sets based on the target region of each parasite. A comprehensive literature review and a self-creation process were undertaken to select thirteen detection method sets for *A. simplex* and *P. decipiens*. The sensitivity of these sets was then evaluated by comparing the $C_q$ values from extracted DNA. The concentrations of six primer/probe sets detected through the screening process were then compared to optimize the test method. The resultant optimized method demonstrated a limit of detection of 0.0019 ng/μL for *A. simplex* and 0.0001 ng/μL for *P. decipiens*. The specificity test also confirmed that there was no cross-activity with the five parasite samples and the three types of anisakids plasmid DNA. This study would contribute development of a rapid detection method for anisakidosis, providing a foundation for proactive responses to food poisoning outbreaks.

## Introduction

Anisakidosis, a zoonotic disease that occurs worldwide, is caused by the larvae of nematodes that parasitize the stomachs of marine mammals, such as whales,

---

**Data availability statement:** All relevant data are within the manuscript.

**Funding:** This research was supported by the Ministry of Food and Drug Safety as part of the Institute's R&D project (23191MFDS013 to I.S.J.). The funders had no role in study design, data collection and analysis, decision to publish, or preparation of the manuscript

**Competing interests:** The authors have declared that no competing interests exist.

**Abbreviation:** mtDNA, mitochondrial DNA; ITS, Internal transcribed spacer; qPCR, Quantitative real-time polymerase chain reaction

dolphins, and seals. Anisakiasis is a parasitic infectious disease transmitted to humans by consuming infected raw fish [1].

Anisakidae family`s thousands of eggs in the marine mammals body, which are then shed in feces. After floating in water for a while, the larvae develop into the third stage larvae (L3) in crustaceans of fish that have consumed crustaceans. These crustaceans are then predated upon by fish or squid, where the larvae mature into adults in the intestines. These L3-infected fish or cephalopods are then ingested raw and infect humans, causing gastrointestinal illnesses [2]. Anisakiasis refers to infection with roundworms and is accompanied by symptoms such as vomiting, chills, and intense pain. IgE-mediated allergic reactions have also been reported [3,4].

The consumption of anisakid larvae generally results in a self-limiting infection that sensitizes individuals to Anisakis allergens. Japan has the highest rate of Anisakis infections among humans, with infected sushi and sashimi—traditional dishes featuring raw fish—being major sources of these infections. Annually, there are between 2,000 and 3,000 reported cases of anisakiasis. The globalization of local cuisines has also led to an increase in the number of people consuming raw fish [3].

Anisakis is widely distributed worldwide, including oceans, with Korea being a key endemic region, along with Japan, the Netherlands, Spain, and Germany. In particular, Japan has experienced a continuous outbreak of whale roundworm disease caused mainly by *Anisakis simplex* and *A. pegreffi* because of the frequent consumption of raw fish [5–7]. In Korea, cases of infection have been consistently reported thus, its global prevalence can be confirmed [8–10].

The family Anisakidae consists of the genera *Anisakis*, *Pseudoterranova,* and *Contracaecum*, and the larvae can be morphologically classified according to the Anisakis species [11]. The most commonly infected species is the Anisakis genus, followed by *Pseudoterranova decipiens*, and rarely *Contracaecum* and *Hysterothylacium* [12]. Recently, it was reported that *A. simplex* is associated with a higher prevalence of hypersensitivity in frozen fish [13]. Meanwhile, the infection rate of *P. decipiens* is correlated with an increase in the number of marine mammals in this area [14].

Anisakidae family are typically identified using traditional morphological identification; however, achieving accurate genus identification is often challenging using the naked eye [9]. Therefore, a molecular approach with high accuracy is required to address this challenge effectively. For the detection of Anisakidae larva, molecular biological methods such as restriction fragment length polymorphism (RFLP), recombinase polymerase amplification (RPA) and loop-mediated isothermal amplification (LAMP) assay have been investigated [15–20].

Therefore, there is a need for specific detection methods which can be used for the confirmation of *A. simplex* and *P. decipiens* in seafood. In this study, we aimed to develop a species-specific TaqMan-based quantitative-polymerase chain reaction (qPCR) method for detecting mtDNA and ITS regions, as well as validate each parasite test method to urgently address food poisoning incidents, such as anisakidosis.

## Materials and methods

### Specimen preparation

**DNA extraction.** Anisakis and Pseudoterranova larvae isolated from marine mammals were provided by the Cetacean Research Institute, National Institute of Fisheries Science, and stored in 100% ethanol until DNA extraction at room temperature. DNA extraction was performed using the DNeasy Blood & Tissue kit (Qiagen) following the steps for the Purification of Total DNA from Animal tissues (spin column protocol). DNA was extracted by cutting a 1-cm section of the body of a single *A. simplex* & *P. decipiens* larva, and DNA quality was checked at a 260/280 nm ratio using a Nanodrop (Thermo Fisher Scientific, USA). DNA extracts were stored at -20 °C until use.

**Synthesis of positive control.** Positive controls were designed for the target sites of the final *A. simplex* and *P. decipiens* gene detection methods. A portion of the template genome sequence, including the detection site considered in the primer design for each parasite, was used as the positive control. The positive control gene was synthesized by Bioneer (Daejeon, Korea).

### Design of primers and probes

Before constructing the primer and probe sets for Anisakis & Pseudoterranova detection, we collected and analyzed research data on domestic and foreign parasite gene detection methods and cataloged the primers/probes for the parasite [21–26].

Based on the unique target regions that could be identified, we focused on a qPCR detection method using TaqMan probes. Among the initially selected primer/probe sets, the detection region was checked using BLAST (https://blast.ncbi.nlm.nih.gov/Blast.cgi), and sets that were not cross-detected were secondarily selected.

In addition to those obtained from the literature [21–26], we designed primers/probes for verification. The sequences of *A. simplex* and *P. decipiens* mtDNA cox2 obtained from the National Center for Biotechnology Information were aligned using the MEGA 11 program [27]. The most common sequence resulting from the alignment was used as the template sequence (*A. simplex* (KC810002.1), *P. decipiens* (OK338708)) and subjected to Genscript (https://www.genscript.com/) (optimal primer size: 20 bp, optimal primer Tm: 60.0 °C, optimal product size: 200 bp), and IDT(https://sg.idtdna.com/) (primer size: 17–30 bp, primer Tm: 59–65 °C, primer GC content: 35–65%, product size: 75–150 bp) using the homepage services of Genscript (optimal primer size: 20 bp, optimal primer Tm: 60). SnapGene™ 1.1.3 program was used to check the position of the primer/probe set to the template sequence. All the primers were synthesized by Genotech (Daejeon, Korea) and all probes were synthesized by Bioneer (Daejeon, Korea).

### qPCR validation

**Adjustment of qPCR methods.** The qPCR mixtures for each method were Taqman Universal Master mix II with UNG (Applied Biosystems, USA), which was configured to the following concentrations: 2 µL of DNA (*A. simplex*: 7.4 ng/µL, *P. decipiens*: 10.7 ng/µL), 10 µL of 2xUniversal mater mix, 400 and 500 nM of primers, and 200 and 250 nM of probe, and distilled water was added to attain a total volume of 20 µL.

qPCR amplification was performed using the 7500 Real-Time PCR System (Applied Biosystems, USA). The thermocycle conditions were adapted from the respective reference methods [28]. In-house thermocycle conditions were as follows: UNG activation at 50 °C for 2 min, polymerase activation at 95 °C for 10 min, denaturation at 95 °C for 15 s for 40 cycles, and annealing and extension at 57–61 °C for 1 min. Each qPCR sample was analyzed in duplicate with no-template control.

For developing the detection method, the MIQE guidelines was used to strengthen methodological approach in experiments. The checklist was summarized in the S1 Table.

**Specificity and limit of detection (LOD) analysis.** To verify the specificity of the finalized *A. simplex* and *P. decipiens* gene detection test methods, eight other species parasites(*Clonorchis sinensis*, *Toxoplasma gondii*, *Metagonimus yokogawai*, *Gymnophalodides seoi*, *Taenia solium*, *Anisakis physeteris*, *Contracaecum osculatum*, and *Pseudoterranova azarasi*) and two larvae(*A. simplex* and *P. decipiens*) were used to confirm non-specific detection. Samples positive for the nine species parasites were collected from the Parasite Laboratory of the Korea Association of Health Promotion. Nucleic acids of the five identified parasites (*C. sinensis*, *T. gondii*, *M. yokogawai*, *G. seoi,* and *T. solium*) were extracted. Plasmid DNA from *A. physeteris*, *C. osculatum*, and *P. azarasi* were used as templates. The specificity of the selected samples was verified using qPCR.

To compare the LOD with the selected primers and probe sets, *A. simplex* and *P. decipiens* DNA products underwent 1/10-fold stepwise dilution and were analyzed across a concentration range of $10^0$–$10^{10}$ copies/μL.

**PCR product verification.** Sanger sequencing was performed with the finalized primer/probe sets. The amplicons were subjected to PCR purification using the AccuPrep® PCR/Gel Purification Kit (Bioneer, Daejeon, Korea). This was followed by a cycling step with the BigDye Terminator kit v. 3.1 (Applied Biosystems, USA) and cleaned up with the BigDye XTerminator v. 3.1 (Applied Biosystems, USA). The cleaned products were sequenced by 3500 Genetic Analyzer (Applied Biosystems, USA).

Sequenced data were aligned to 9 nucleotide sequences which is registered in NCBI and phylogenetic analysis was performed MEGA 11, using the Maximum Likelihood method and Tamura-Nei model [29].

**Inter-examinator validation.** Inter-examiner validation of the method was performed by two investigators in our laboratory once daily for a total of 3 days. The DNA extracted from the anisakids was diluted $10^{-3}$-fold (1xLOD) and 5x $10^{-3}$-fold (5xLOD) for *A. simplex* and $10^{-5}$-fold (1xLOD) and 5x$10^{-5}$-fold (5xLOD) for *P. decipiens*. qPCR was then performed using the diluted DNA as templates together with the final detection method selected for this study. All experiments were performed in triplicate.

## Results and discussion

A literature search for Taqman-based qPCR gene detection in anisakid (Table 1) revealed six studies concerning *A. simplex* and one for *P. decipiens* [2,22–25,30]. In addition, primer/probe sets were designed based on the sequences of *A. simplex* (KC810002.1) and *P. decipiens* (OK338708): one for *A. simplex* and five for *P. decipiens* (Table 2). In total, thirteen sets of primers/probes were selected for the comparison between *A. simplex* and *P. decipiens* (Tables 1, 2).

The positive control was reliably amplified within the expected range of Ct values in all experimental replicates. The Ct value was found to be 13.83 ± 0.01 for *A. simplex* and 13.84 ± 0.12 for *P. decipiens*, demonstrating the high reproducibility and consistency of the experiment.

### Sensitivity of primers and probes for the identification of anisakids and qPCR setup

Comparative experiments were conducted to confirm the detection of Anisakis using the selected reference and constructed test methods.

The qPCR conditions used the qPCR cycling conditions of the selected reference and the parasite gene detection method in the Guidance Documents of Food Poisoning Test [28] (Table 3).

The results showed that three instances of *A. simplex* and five instances of *P. decipiens* were detected (Fig 1). Therefore, we selected each primer/probe set that confirmed the detection of Anisakis during the test according to the primer set.

Using the reference concentration (400 nM:200 nM) listed in the Guidance Documents of Food Poisoning [28], the selected sets were compared using different primer and probe concentrations. The method with the most stable Ct value (< 30) was selected based on a comparison between methods using actual extracted DNA [22] (Table 3 and Fig 2). The primer:probe concentration combinations (400 nM:200 nM, 400 nM:250 nM, 500 nM:200 nM, and 500 nM:250 nM) were applied as PCR conditions for each set. We further optimized the detection method by analyzing qPCR assays with

**Table 1. Selected sets of primers and probes for detection based on references.**

| Type | Primer name | Target | Sequence (5′-3′)[a] | Product size (bp) | Annealing Temperature (°C) | Reference |
|---|---|---|---|---|---|---|
| *A. simplex complex* | AS1_F | ITS | GACTGTGAAGCATTCGGCAAG | 161 | 62 | [28] |
| | AS1_R | | ACCACCAAAAGCACCTCACC | | | |
| | AS1_P | | FAM-CTATCATGGACAATATGACGAGCGGTTC-TAMRA | | | |
| | ASs1_F | | TTTTGGCTGCTAATCATCATTGA | 76 | 60 | [21] |
| | ASs1_R | | CCACCTAGCGTGGCTCATTAA | | | |
| | ASs1_P | | VIC-TAGCTTAAGGCAGAGTTG-MGB | | | |
| | ASs5_F | ITS1 | GAACAACGGTGACCAATTTGG | 56 | 60 | [22] |
| | ASs5_R | | GACGGTCCAGGCAGAAGCT | | | |
| | ASs5_P | | VIC-TACGCCGTATCTAGCTTC-MGB | | | |
| | ASs2_F | COX2 | CTTTAATTTTGGTTGCTCAGAT | 217 | 60 | [23] |
| | ASs2_R | | CGATTATCAACCTCCAAAAG | | | |
| | ASs2_P | | CY-ATGACCAGTGACTTTCACAGTCAAAT-BBQ | | | |
| | ASs3_F | | AGTAAGAAGATTGAATATCAGTTTGGTGA | 61 | 60 | [24] |
| | ASs3_R | | AAGTAAACTCAAAGAAGGCACCATC | | | |
| | ASs3_P | | FAM-TTCCTACTTTAATTTTGGTTGCTC-MGB | | | |
| *A. simplex complex & P. decipiens* | AS2_F | COX1 | GGKCYATTAAYTYTATRACWACTAC | 176 | 52 | [25] |
| | AS2_R | | AAAGAWGTATTMARRTTACGRTCVG | | | |
| | AS2_P | | FAM-TCTATTTCTTTGGARCAYA-TAMRA | | | |

[a]Mixed based; R = A + G, Y = C + T, S = G + C, W = A + T, K = G + T, M = A + C, B = C + G + T, D = A + G + T, H = A + C + T, V = A + C + G, N = any base.

**Table 2. Selected sets of newly designed primers and probes for the detection of *A. simplex* and *P. decipiens*.**

| Type | Primer name | Target | Sequence (5′-3′) | Product size (bp) | Annealing Temperature (°C) |
|---|---|---|---|---|---|
| *A. simplex complex* | ASS pair1_F | COX2 | CCGGGCTTAGAGTTTGA | 103 | 57 |
| | ASS pair1_R | | CACAAGGGACAACACAACGA | | |
| | ASS pair1_P | | FAM-ACGAGGCTCGCCTAGCTCCA-TAMRA | | |
| *P. decipiens* | PD pair1_F | | AGAGACATTCCTGGTTTGGAGT | 172 | 58 |
| | PD pair1_R | | CTAGGCAAAGCCCAAGAA | | |
| | PD pair1_P | | FAM-ACGTGGCTCTCCCAATTCCAACTGA-TAMRA | | |
| | PD pair1 J_F | | TCCTGGTTTGGAGTTTGATTCT | 101 | 59 |
| | PD pair1 J_R | | CACAAGGAACAACACAACGATTA | | |
| | PD pair1 J_P | | FAM-TCAACCTCTAACAAACGTGGCTCCC-TAMRA | | |
| | PD pair2 J_F | | CGTTGTGTTGTTCCTTGTGATAC | 111 | 60 |
| | PD pair2 J_R | | ACCTCTCATAGCATCCAACTTAAT | | |
| | PD pair2 J_P | | FAM-TTCATTCTTGAGCTTTGCCCAGGA –TAMRA | | |
| | PD pair3 J_F | | GGGAGCCACGTTTGTTAGA | 107 | 59 |
| | PD pair3 J_R | | GGGCAAAGCTCAAGAATGAATTA | | |
| | PD pair3 J_P | | FAM-TCACAAGGAACAACACAACGATTATCAACC-TAMRA | | |
| | PD pair4 J_F | | GCCACGTTTGTTAGAGGTTGATA | 106 | 60 |
| | PD pair4 J_R | | CCTGGGCAAAGCTCAAGAA | | |
| | PD pair4 J_P | | FAM-TCGTTGTGTTGTTCCTTGTGATACT-TAMRA | | |
| | PD pair5 J_F | | GGGATATTCCTGGTTTGGAGTT | 121 | 61 |
| | PD pair5 J_R | | CGAATATTAGTATCACAAGGAACAACAC | | |
| | PD pair5 J_P | | FAM-TCAACCTCTAACAAACGTGGCTCCC-TAMRA | | |

**Table 3. Screening of primer/probe sets for each *A. simplex* & *P. decipiens* sample.**

| Type | Region | Primer | Product size (bp) | Ct value |
|---|---|---|---|---|
| *A. simplex* | ITS | AS1 | 161 | ND |
| | | ASS1 | 76 | ND |
| | ITS1 | ASS5 | 56 | 38.53 |
| | COI | AS2 | 178 | ND |
| | COX2 | ASS2 | 217 | ND |
| | | ASS3 | 96 | 23.77 |
| | | ASS pair1 | 103 | 39.79 |
| *P. decipiens* | COX2 | PD pair1 | 172 | ND |
| | | PD pair1 J | 101 | 19.77 |
| | | PD pair2 J | 111 | 24.52 |
| | | PD pair3 J | 107 | 21.75 |
| | | PD pair4 J | 106 | 22.76 |
| | | PD pair5 J | 121 | 20.62 |

ND, Not detected.

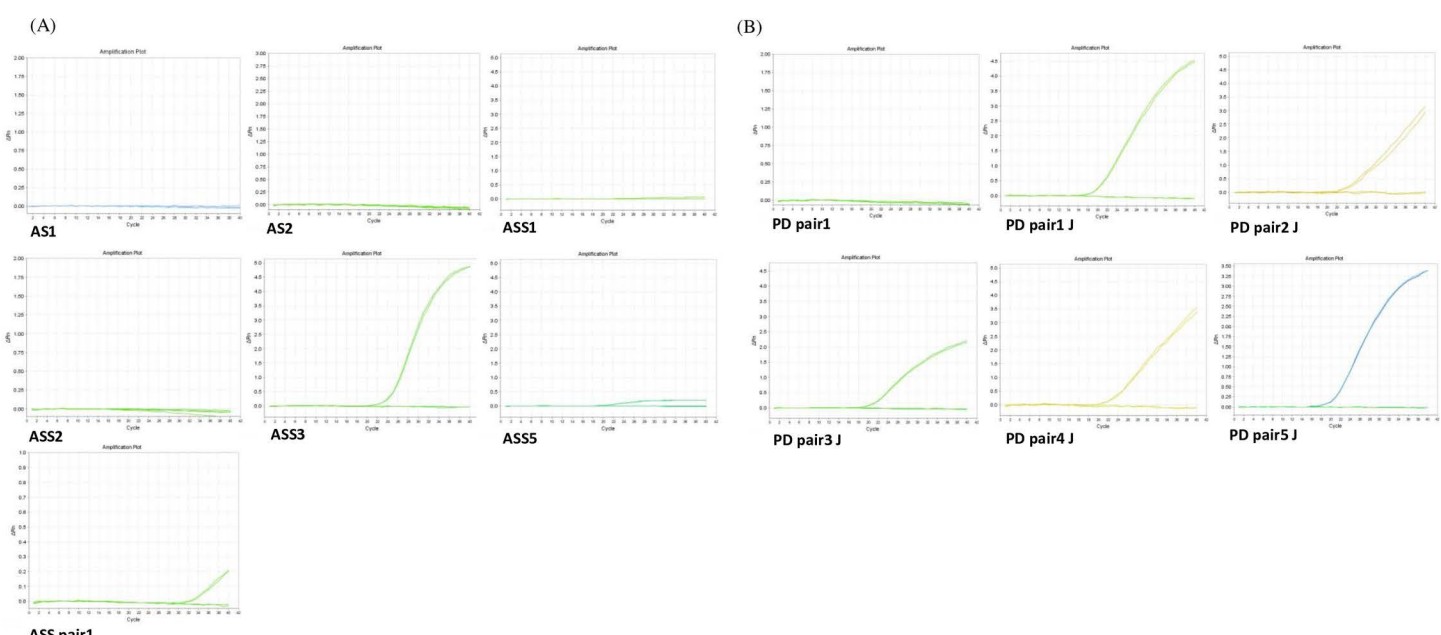

**Fig 1. Amplification curves based on various primer types of each species.** (A) *A. simplex* and (B) *P. decipiens* samples were used for screening primer/probe sets.

different primer/probe set concentration combinations. The final concentrations selected were 400 nM/250 nM for *A. simplex* and 400 nM/200 nM for *P. decipiens* (Table 4 and Fig 2).

## Specificity of optimal qPCR method

The detection regions of the selected primer/probe sets were first checked for overlap with those of other Anisakis species and genera using BLAST, and their specificity was confirmed via qPCR using actual parasite samples from

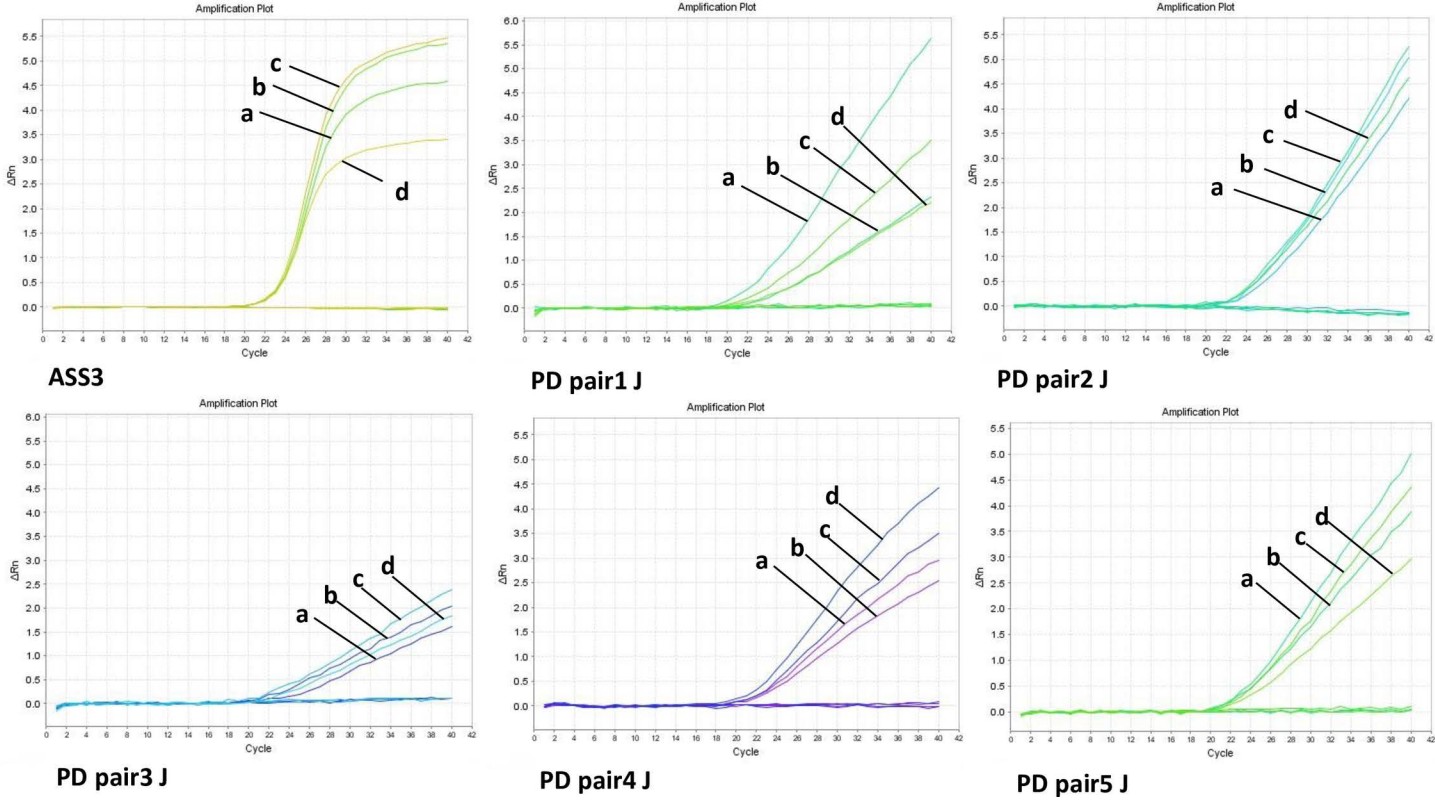

**Fig 2. Comparison of concentration corresponding to various primer sets.** Amplification curves of different concentration combination of primer:probe (nM) sets; a = 400:200, b = 400:250, c = 500:200, d = 500:250.

**Table 4. Combination of primers and probes with different concentration for comparison of Ct values using qPCR.**

| Type | Primer | Primers:Probe (nM:nM) | | | |
|---|---|---|---|---|---|
| | | 400:200 | 400:250 | 500:200 | 500:250 |
| *A. simplex* | ASS3 | 22.643±0.139 | 22.421±0.062 | 22.433±0.122 | 22.499±0.162 |
| *P. decipiens* | PD pair1 J | 20.496±0.024 | 23.543±0.25 | 22.735±1.322 | 23.181±0.084 |
| | PD pair2 J | 22.674±1.603 | 23.463±0.125 | 23.218±0.299 | 23.302±0.202 |
| | PD pair3 J | 24.204±1.258 | 22.799±0.095 | 22.269±0.716 | 23.143±0.026 |
| | PD pair4 J | 22.128±0.253 | 22.254±0.154 | 21.789±0.209 | 21.147±0.381 |
| | PD pair5 J | 21.626±0.091 | 21.911±0.172 | 22.109±0.108 | 22.846±0.179 |

The data are represented as mean±standard deviation (SD).

various species. As a result, *A. simplex* and *P. decipiens* each showed one independent amplification curve, confirming that there was no cross-reaction between species. Gene amplification was not detected in parasites of other species or anisakids of the same species. The qPCR amplification curves showed that only the corresponding anisakids were amplified (Fig 3). This confirmed that the selected detection method was valid for the specific detection of certain anisakids.

The reliability of the specifically detected PCR product was finally secured through the sequencing process. According to Fig 4, the sequencing results for the two species confirmed that they are *A. simplex* an d *P. decipiens*.

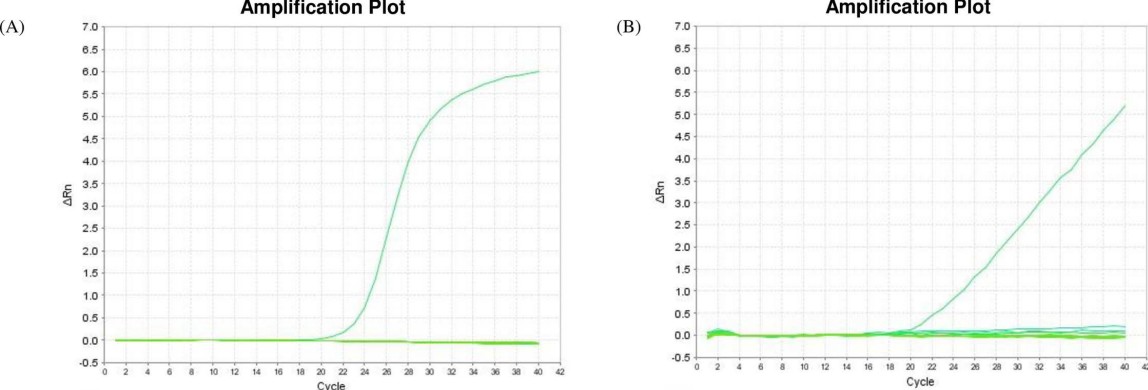

**Fig 3. Specificity for optimized and qPCR methods to detect anisakidae family.** The amplification curves of anisakidae family were verified using the other parasites. (A) *A. simplex* (B) *P. decipiens.*

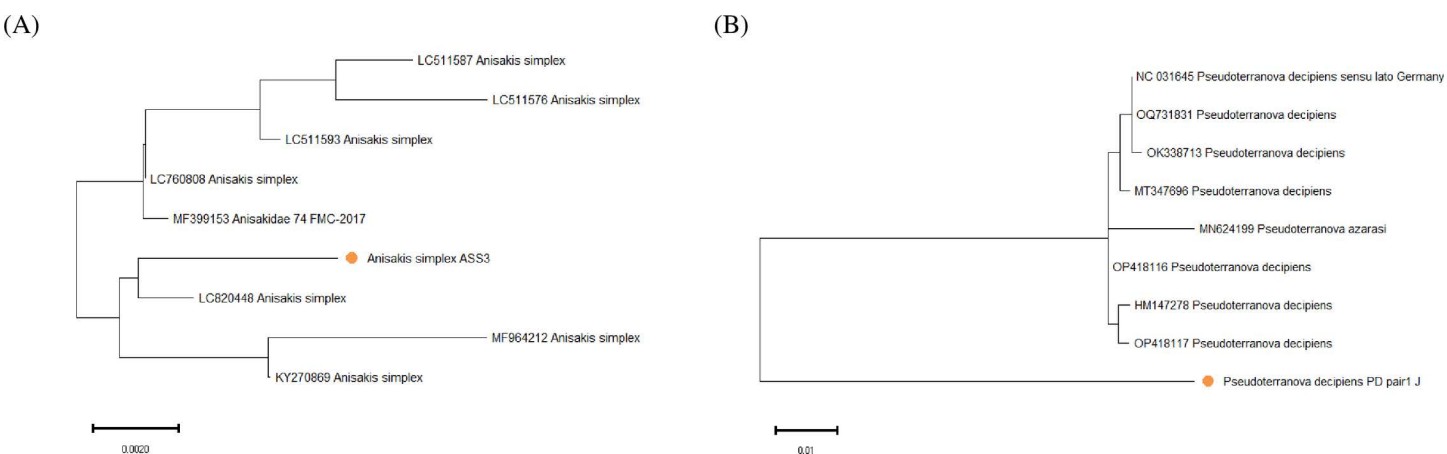

**Fig 4. Phylogenetic tree analysis of (A) *A. simplex* and (B) *P. decipiens.*** The evolutionary analyses were conducted in MEGA11 and inferred by using the Maximum Likelihood method and Tamura-Nei model [27]. Circled in orange is the analyzed samples.

## LOD

The extracted DNA was diluted 1/10-fold stepwise and used as a template. The detection limits were confirmed as $1.93 \times 10^7$ copies/µL for *A. simplex* and $9.67 \times 10^5$ copies/µL for *P. decipiens*. In addition, the amplification efficiency (E) was 91% for *A. simplex* and 110% for *P. decipiens*, and their corresponding correlation coefficients ($R^2$) were 0.9998 and 0.9949, respectively (Fig 5). This strong linear relationship ($R^2 > 0.99$) indicates that the qPCR assay is suitable for Anisakis detection.

Based on the optimized test method, the LODs were 0.0019 and 0.0001 ng, with high sensitivity for *A. simplex* and *P. decipiens*, respectively. This obtained sensitivity is consistent with other studies [22,23,25,31], where LODs ranged from 0.3 to 2 pg, with *P. decipiens* detected down to 0.1 pg. Additionally, Ct values were also within acceptable tolerances in cross-validation.

## Cross-examiner validation

To ensure the reliability of the anisakids gene test method selected in this study, cross-validation was conducted in triplicate between the two investigators. As a result of detection by qPCR, the CV values of 1xLOD and 5xLOD of *A. simplex* gene

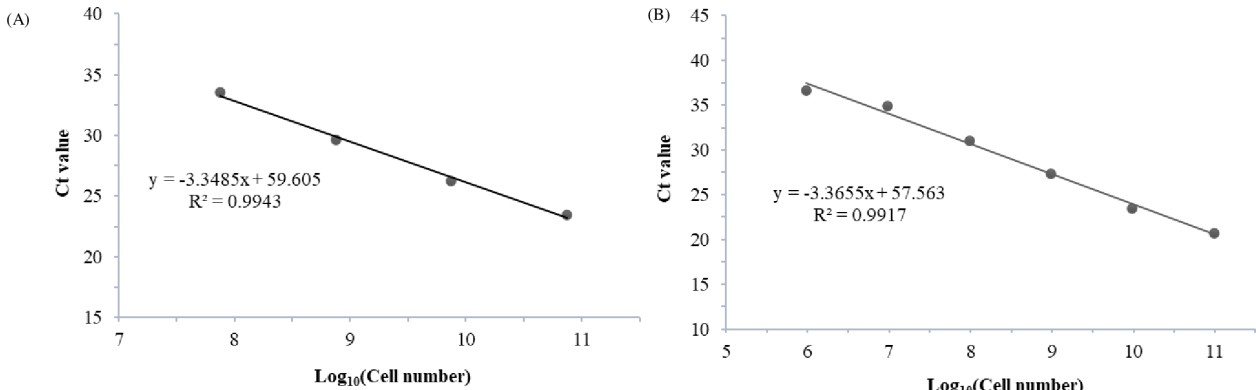

**Fig 5. Standard curve of the Ct value and logarithm of *A. simplex* and *P. decipiens* DNA amounts.** Linearity test of 10-fold diluted series of (A) *A. simplex* and (B) *P. decipiens* DNA samples.

amplification products of Investigator A were 0.50% and 0.26%, respectively, and 1.81% and 0.76%, respectively, for *P. decipiens*. Meanwhile, Investigator B identified CV values of 0.68% (1xLOD) and 0.28% (5xLOD) for *A. simplex* and 1.61% (1xLOD) and 2.24% (5xLOD) for *P. decipiens*, with each data point representing the average of the triplicate (Table 5 and Fig 6).

In both experiments, no amplification reaction was observed in the negative control (NC), indicating that the PCR was working properly. The CV values of 1xLOD and 5xLOD were less than 4%; therefore, they were judged repeatable and reproducible.

Our study was aimed to analyze potential foodborne pathogens related to food consumption habits and climate change using qPCR methods. Accordingly, we aimed to investigate and compare genetic testing methods for parasites that may cause acute or chronic infections in food items, which are not frequently reported in South Korea but are a concern for the future. This was done by reviewing domestic and international research papers to identify the optimal genetic testing method.

Among other molecular biology techniques, qPCR has been applied as a rapid detection method to a variety of microorganisms, including bacteria, viruses, and parasites due to its speed, accuracy, and sensitivity [26,32–34].

In particular, it was believed that qPCR would yield faster results compared to conventional PCR methods, so this study was conducted by using this approach. Additionally, we considered the PCR composition methods utilized by the Korea Food and Drug Administration for genetic testing of parasites during foodborne illness investigations.

Based on this, it is expected that accurate, rapid, and cost-effective identification of anisakids found in foods, such as fish and seafood, will be possible. In addition, a method for detecting stable genes other than mtDNA sequences needs to be developed. Quantitative experiments should also be conducted based on qualitative experiments that determine the presence of parasites.

**Table 5. Inner-cross checking of anisakids through researcher A and B using qPCR.**

| | | Mean | | SD | | CV | |
|---|---|---|---|---|---|---|---|
| | | A | B | A | B | A | B |
| *A. simplex* | 1xLOD | 32.62 | 32.44 | 0.16 | 0.22 | 0.50 | 0.68 |
| | 5xLOD | 30.24 | 30.09 | 0.08 | 0.08 | 0.26 | 0.28 |
| *P. decipiens* | 1xLOD | 38.42 | 37.50 | 0.69 | 0.60 | 1.81 | 1.61 |
| | 5xLOD | 36.43 | 36.55 | 0.22 | 0.82 | 0.76 | 2.24 |

CV; SD/Mean.

(A)

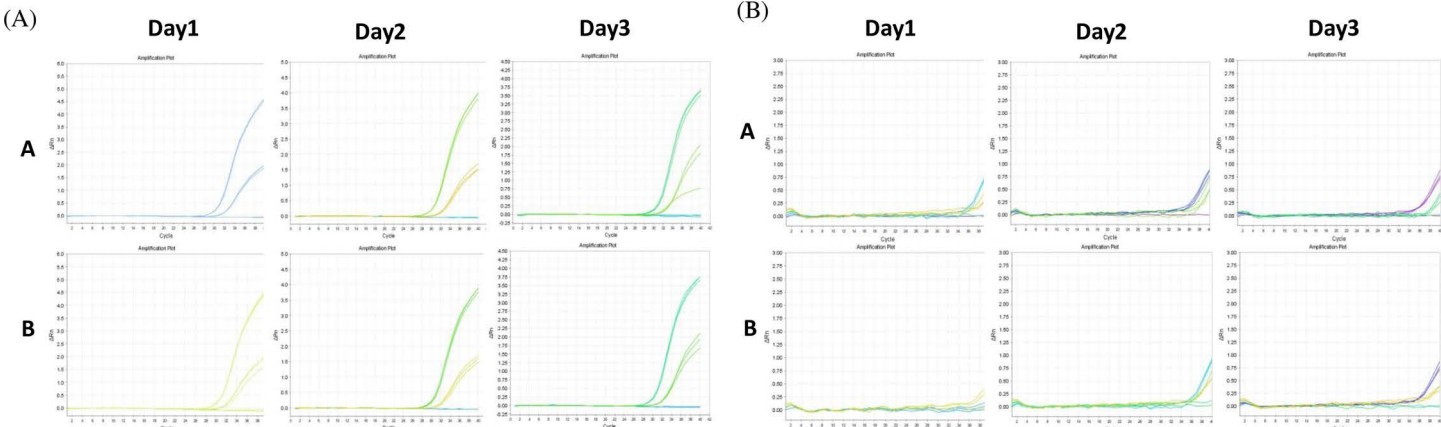

(B)

**Fig 6. Validation of cross-reaction with between investigators.** Cross-validation through internal researchers (A and B) was conducted based on final selected detection methods. (A) *A. simplex* and (B) *P. decipiens* genes were amplified under the same conditions for three days. Among the two types of amplification curves, the relatively higher amplification curve represents 5 x LOD, while the other represents 1 x LOD.

## Conclusion

Anisakids are typically identified using traditional morphological identification; however, achieving accurate genus identification is often challenging using the naked eye [9]. Therefore, a genetic approach with high accuracy is required to address this challenge effectively.

Recently, qPCR methods targeting various regions have been developed for the detection of Anisakis [2,17,22,24,25,30,35]. In this study, we designed our own primer/probe set targeting the cytochrome c oxidase subunit II (COX2) region to develop a more stable detection method in several mitochondrial genomes.

In summary, this study developed an effective qPCR detection method for *A. simplex* and *P. decipiens* that has the potential to be standardized. As a result of comparing genetic optimized methods for detecting two types of parasites, it was able to identify a method with high sensitivity and specificity in the laboratory. Consequently, the sequenced qPCR products of the selected primer/probe sets confirmed that each resulted in the target parasite by phylogenetic analysis. This method is expected to assist in the future identification of these two parasites using genetic amplification techniques.

## Supporting information

**S1 Table. Partial list of MIQE checklists [1].** From the list, select only those that are applicable to this paper. All essential information (E) must be submitted with the manuscript. Desirable information (D) should be submitted if available. If using primers obtained from RTPrimerDB, information on qPCR target, oligonucleotides, protocols and validation is available from that source. FFPE, formalin-fixed, paraffin-embedded; RIN, RNA integrity number; RQI, RNA quality indicator; GSP, gene-specific priming; dNTP, deoxynucleoside triphosphate.
(DOCX)

## Acknowledgments

We would like to thank the Cetacean Research Institute of the National Institute of Fisheries Science for providing *A. simplex* and *P. decipiens* specimens.

## Author contributions

**Conceptualization:** Doo Won Seo.

**Data curation:** Mi-Gyeong Kim.

**Investigation:** Mi-Gyeong Kim, Min Ji Hong.

**Methodology:** Mi-Gyeong Kim, Min Ji Hong, Doo Won Seo.

**Supervision:** Hyun Mi Jung, Hyun-Ja Han, Seung Hwan Kim, Insun Joo.

**Visualization:** Mi-Gyeong Kim.

**Writing – original draft:** Mi-Gyeong Kim.

**Writing – review & editing:** Mi-Gyeong Kim, Doo Won Seo, Hyun Mi Jung, Hyun-Ja Han.

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
