## [Decision Letter · Decision Letter 0]

8 Sep 2024

PONE-D-24-33370Comparative study of Taqman-based real-time PCR assay for the detection of Anisakis simplex and Pseudoterranova decipiensPLOS ONE

Dear Dr. Kim,

Thank you for submitting your manuscript to PLOS ONE. After careful consideration, we feel that it has merit but does not fully meet PLOS ONE’s publication criteria as it currently stands. Therefore, we invite you to submit a revised version of the manuscript that addresses the points raised during the review process.

The authors need to have clear and in depth/thorough elaboration of all sections of the manuscript, including clear introduction of addressed problem, material and methods, including statistical analysis, results and their respective discussions. The authors need to further elaborate novelty of their detection method as compared to what has already been published by other researchers.

We look forward to receiving your revised manuscript.

Kind regards,

Elingarami Sauli, PhD

Academic Editor

PLOS ONE

Journal Requirements:

"This study was supported by a grant [No. 23191MFDS013] from the Ministry of Food and Drug Safety (MFDS). The findings and conclusions of this study are our own, and do not necessarily represent the views of the Ministry of Food and Drug Safety. We would also like to thank the Cetacean Research Institute of the National Institute of Fisheries Science for providing A. simplex and P. decipiens specimens."

"This study was supported by the Ministry of Food and Drug Safety as part of the Ministry's R&D project"

 "This study was supported by the Ministry of Food and Drug Safety as part of the Ministry's R&D project."

Reviewers' comments:

Reviewer's Responses to Questions

**Comments to the Author**

1. Is the manuscript technically sound, and do the data support the conclusions?

Reviewer #1: Partly

Reviewer #2: Partly

Reviewer #3: Partly

Reviewer #4: No

2. Has the statistical analysis been performed appropriately and rigorously? 

Reviewer #1: N/A

Reviewer #2: Yes

Reviewer #3: No

Reviewer #4: No

3. Have the authors made all data underlying the findings in their manuscript fully available?

Reviewer #1: Yes

Reviewer #2: No

Reviewer #3: No

Reviewer #4: No

4. Is the manuscript presented in an intelligible fashion and written in standard English?

Reviewer #1: Yes

Reviewer #2: Yes

Reviewer #3: Yes

Reviewer #4: No

5. Review Comments to the Author

Reviewer #1: The MS presented to me for review is well attempted effort to develop a new Taqman-based real-time PCR assay for the detection of Anisakis simplex and Pseudoterranova decipiens. However, introduction should have included extent of the problem due to raw fish eating and population susceptible. Apart from this, the development is based on DNA extraction from Anisakis larvae provided apart from other larvae and parasitic species obtained. I would have wish to see, if they could have further tested the diagnostic probe in suspected food samples likely to be consumed by human. Rest others seems Ok to me and few marked in MS.

Reviewer #2: The manuscript (PONE-D-24-33370) entitled "Comparative study of Taqman-based real-time PCR assay for the detection of Anisakis simplex and Pseudoterranova decipiens is a interesting article that addresses the important issue of obtaining method detection was that are critical for the evaluation to control present of Anisakid in food.

Parasites family Anisakidae are fish-borne parasites that are responsible for a large number of human infections and allergic reactions around the world. World health organizations and food safety authorities aim to control and prevent this emerging health problem. Specially Anisakis simplex and Pseudoterranova decipiens are main dangerous for human health. A rapid and sensitive detection strategy (DNA, protein) can provide authorities with an effective and fast method to guarantee consumer safety, therefore research and methods are particularly important.

Comments:

Line 36 Anisakiasis, is a name of diseases cause only nematode parasite belong to genus Anisakis

Line 37: “third-instar” is description of insect larval stage!!!!!! https://doi.org/10.1016/B978-0-12-820359-0.00002-5.

Line 31 and 51, 54 Anisakis, precise meaning - genus? mistake – Anisakids? And many times, the same in whole manuscript!!!!

Line 61 remove” infected” repetition

Line 94,98 only larvae of Anisakis not Pseudoterranova???

Line 123, how reference methods? sspecific papers?

Line 124, molecular material, template is DNA. Where does cDNA synthesis come from? The material, template was RNA?

What are the stages of PCR? Basic knowledge, denaturation, annealing, extension……

Line 129, 130 etc. What 9 parasites? What 2 larvae?

Line 146, 3 d? 3 days? 72 hours?

Line 164, What ascaridoid species?

Line 220, cell numbers? cell of plate?

In the "Results and discussion" in my opinion it is necessary to compare and clearly present the protocol for using the method sensitivity demonstrated for the authors!

Little Explanation:

Anisakis – genus

Anisakiosis/Anisakiasis - disease cause for Anisakis genus

Anisakidae family (Anisakis, Pseudoteranova, Contraceucum)

Anisakidosis disease cause for family Anisakidae (Anisakis, Pseudoterranova, Contraceucum)

Anisakids, English colloquial, customary, non-taxonomic!!!using in papers

fish-borne parasites belong to family Anisakidae

Detection DNA from larvae, have there been any attempts to detect DNA of parasite from tissue of fish?

As the authors write, there are many papers (2, 16, 18) on the detection of Nematodes belong to Anisakidae, there is a lack of discussion with other authors, and above all, there is no demonstration of higher sensitivity of the method validation by the authors, a protocol for further research, and use in detection in food.

How aim of this work?

The paper has many mistakes substantive, knowledge of taxonomy, nomenclature,

deficiencies in the methodology and ambiguities in the description of the method.

The work is scientifically sound and might to consideration for publication after thorough improvement. The experimental design is correct but not Research Paper maybe as a “Submissions describing methods and validation- protocol?”

Reviewer #3: Manuscript Number: PONE-D-24-33370

Comparative study of Taqman-based real-time PCR assay for the detection of Anisakis simplex and Pseudoterranova decipiens

Authors: Mi-Gyeong Kim, Min Ji Hong, Doo Won Seo, Hyun Mi Jung, Hyun-Ja Han, Seung Hwan Kim, Insun Joo

General comments:

This manuscript presents a comparative study of a TaqMan-based real-time PCR assay for the detection of two heteroxenous parasites, Anisakis simplex and Pseudoterranova decipiens. These parasites are among the most significant biological hazards in seafood products, causing Anisakiasis in humans. Accurate detection is therefore crucial for ensuring food safety and protecting public health.

However, the manuscript requires significant improvements before it can be considered for publication. Specifically, the writing needs enhancement, particularly in the Results and Discussion sections. It is important to clearly differentiate between the findings and provide a detailed discussion of each result. Expanding the Results and Discussion sections will make the manuscript more robust and suitable for publication. Additionally, the Materials and Methods section should be elaborated to ensure that readers can easily understand the experimental procedures.

For the qPCR section, it is recommended to adhere to the Minimum Information for Publication of Quantitative Real-Time PCR Experiments (MIQE) guidelines (Bustin et al., 2009) and use the appropriate terminology to avoid any confusion. Including the MIQE checklist in the supplementary data would also be beneficial.

References

Bustin, S.A., Benes, V., Garson, J.A., Hellemans, J., Huggett, J., Kubista, M., Mueller, R., Nolan, T., Pfaffl, M.W., Shipley, G.L., 2009. The MIQE guidelines: minimum information for publication of quantitative real-time PCR experiments. Clin. Chem. 55, 611–622

Other comment and suggestions: -

Introduction: It is suggested to include previous studies that employed different approaches for detecting Anisakis simplex and Pseudoterranova decipiens, or other members of the Anisakidae family, such as multiplex PCR or alternative primers. Discussing the limitations of these previous studies and the rationale behind designing the current study would enhance the reader's understanding of the study's background and objectives.

Line 99: Clarification is needed regarding whether the research data mentioned here refers to data from previous studies on domestic and foreign parasite genes. If it does, please provide the appropriate citations. Additionally, it would be more informative to specify the types of data reviewed (e.g., sequences, detection protocols) and the criteria used to select relevant methods.

Line 103: The term "not cross-detected" could be made more explicit by specifying the criteria used, such as the specific thresholds or parameters applied in the BLAST search.

Line 105: Please provide citations for the literature referenced in this section.

Line 106-107: Please provide the accession numbers for all taxon sampling sequences. These can be included in the supplementary data.

Line 109: Please provide a citation or link for "Genscript." If available, include the version used as well.

Section Design Primers and Probe: Clarification is needed on whether the designed primers and probes were validated through in silico analysis for primer specificity using the NCBI database.

Line 123: Clarification is needed regarding what the reference methods refer to. Providing the exact references or more details on how these methods were adapted (e.g., changes in cycling conditions, reagent concentrations) would improve transparency and allow others to replicate or build on the work.

Line 126: Clarification is needed on whether the study considered a result valid and if any quality control measures were applied, such as the inclusion of no-template controls or positive controls.

Line 129: Please specify the nine other parasite species and the two larvae used to confirm non-specific detection. Are these species closely related to Anisakis and Pseudoterranova, or were they selected because they are commonly found in similar environments? Explaining the rationale behind the selection of these particular species would be beneficial.

Line 133: Please clarify whether these plasmids contain full-length gene sequences or specific target regions, and whether any controls were used to ensure the accuracy of these plasmid constructs.

Additionally, provide details on how the plasmid DNA from Anisakis physeteris, Contracaecum osculatum, and Pseudoterranova azarasi was synthesized. Was this done using a cloning kit? Specify the type of plasmid DNA used in the study and indicate whether the plasmid was linearized using a restriction enzyme (RE).

Section Synthesis of Positive Control: This section would benefit from additional details. Please elaborate on how the positive control was synthesized. Was it generated using a PCR amplicon, synthetic gene fragments, or PCR synthesis from a plasmid? Additionally, specify which region was selected for the synthesis.

Section Inter-Examiner Validation: Additional details would enhance the clarity and comprehensiveness of this section. Consider the following suggestions:

-Briefly explain the purpose of inter-examiner validation.

-Clarify the exact concentrations of the dilutions used, specifying them in terms of DNA copies per microliter or nanograms per microliter for precision.

-Indicate the number of replications performed.

-Specify the statistical tests used to compare the results between the two investigators, such as interclass correlation coefficients or Bland-Altman analysis, to assess agreement or consistency between examiners.

Provide the rationale for conducting the validation once daily for three days. Was any longer-term validation (e.g., over weeks or months) planned or conducted?

Line 153: Please specify the starting year for the 10-year period mentioned.

Line 154: The citation "(Table 1)" would be more appropriately placed at the end of the statement, following "P. decipiens," as in "P. decipiens (Table 1)..."

Line 155-157: The study's findings should be clearly stated here. It is suggested to rephrase as: "This study designed several sets of primers and probes based on the sequences of..."

Line 157: Clarification is needed regarding the "seven detection methods" mentioned. Are these methods referring to all the primer/probe sets used in the qPCR assay, or are they separate techniques? Please rephrase this sentence to avoid confusion.

Line 158-159: The sentence "The detection regions of the selected primer/probe sets were first checked for overlap with those of other Anisakis species and genera using BLAST..." refers to the in-silico analysis of primer/probe specificity. It would be more useful to present the results of this analysis in the manuscript. Specifically, indicate whether all BLAST hits were on the targeted species, and if there were any hits on non-target species. Provide details on the percentage of hits, as well as the similarity and coverage percentages.

Line 160-161: Although the real-time PCR showed a single independent amplification, this alone does not confirm that the detected amplicon is from the targeted region for the species. To ensure accuracy, it would be better to perform sequencing of the PCR or real-time PCR product to confirm the identity of the amplicon.

Line 178: The term "selected reference" needs clarification. Is it referring to the primers/probes shown in Table 1? Please clearly specify this to help readers better understand the context.

Line 178-179: The term "PCR conditions" may cause confusion. It is suggested to either refer to the MIQE guidelines for appropriate terminology or replace it with "qPCR cycling conditions" for clarity.

Line 180-181: It would be more effective to include the qPCR amplification curve results in this section to provide clearer evidence of the findings.

Line 186-187: Please include the amplification curve as part of the results for this finding.

Line 187-192: The current statement reads more like a description of materials and methods rather than presenting actual results. To improve clarity, rephrase the section by explaining the outcomes shown in Table 4. Additionally, it is recommended to include the amplification curves as part of the result to better illustrate the findings.

Table 3: Please include the primer sequences and their corresponding annealing temperatures in the table.

Line 202: Were the primers/probes tested on wild samples?

Additionally, this section should include some discussion rather than just presenting the results, as this is the Results and Discussion section.

Line 211: It would be beneficial to include the equation for converting copies/µL to ng or cells in the Materials and Methods section. This addition would provide clarity on the calculation process.

Fig 2: The statement in lines 201-215 mentions the detection limit in terms of gene copy number (copies/µL), but Figure 2 shows log10 cell number. Please confirm if the figure label is correct. Additionally, clarify the relationship between gene copy number and cell number. What is the gene copy number per cell?

Line 225: In addition to Table 5, it is crucial to provide the raw data for the amplification curves in the supplementary materials to demonstrate the reliability of these findings.

Line 243-249: The statements in this section are more appropriate for the Introduction. Please consider moving them to the Introduction section.

Reviewer #4: The authors deal with a parasitic nematode of public health importance, Anisakis simplex and Pseudoterranova decipiens. Unfortunately, the paper submitted for review is written in an unreliable manner and raises my doubts.

The attempt made by the authors to produce a Taqman probe for the detection of the above-mentioned nematode species in food was, in my opinion, not carried out in accordance with the current standards of molecular biology. For example, qPCR is a quantitative method, and the authors used different amounts of DNA matrix of the tested species for the analyzes, which is a mistake (line 119).

In addition, the preparation of the primer sequence and probes is not clearly discussed. There is a lack of alignments in the form of a figure with a discussion or at least in the supplement.

At the same time, according to the authors, the analysis of specificity and detection limit was based on nine other parasitic species. I have mentioned 8 other species in the text. Why were these species selected? There are spelling errors in the species names (correct spelling Gymnophalloides seoi).

Why was plasmid DNA from Anisakidae species used?

The manuscript lacks electrophoresis of the products after the PCR reaction, and these products should be sequenced to finally confirm the specificity of the amplification.

Moreover, the manuscript lacks information on statistical analysis, number of tests performed, number of positive and negative results, etc.

Furthermore, as the authors emphasized at the beginning of the paper, there are currently verified methods for the detection of species from the Anisakidae family. This study does not introduce any new detection methods, and it is difficult to verify whether this probe is better than others currently available because the authors did not provide sufficient information and did not discuss the results based on other work.

At the same time, the authors write about identification in food, which has also not been verified. A matrix of infected and non-infected foods should be created and analyzed.

Apart from the methodological part, the introduction is laconic and the sentences are repeated. Much of the information, for example about the development cycle, is incorrect. For example, "predated upon by fish... where the larvae mature into adults in the intestines" in line 65 is complete nonsense.

The manuscript also contains some linguistic errors. In line 77, for example, I think it should read "infectious" and it reads infected, which gives the sentence a completely different meaning. The text should be edited by a native English speaker.

In conclusion, I do not recommend the manuscript for publication in Plos One. The authors should thoroughly revise the manuscript and consider a variety of details before it can be resubmitted. However, it should not be accepted in its current form as this manuscript is not written and presented according to the standards of this journal.

6. PLOS authors have the option to publish the peer review history of their article (what does this mean? ). If published, this will include your full peer review and any attached files.

**Do you want your identity to be public for this peer review?** For information about this choice, including consent withdrawal, please see our Privacy Policy .

Reviewer #1: **Yes: ** Dr Rashmi Rekha Kumari

Reviewer #2: No

Reviewer #3: No

Reviewer #4: No

---

## [Author Response · Author response to Decision Letter 0]

29 Oct 2024

Hello, I am Seo Doo Won, the corresponding author of this paper, from the Microbiology Division of the Korea Food and Drug Administration, conducting research related to food safety management in Korea.

First of all, thank you for providing many professional opinions despite your busy schedule. We are very grateful to the helpful and insightful comments from Journal Requirements, Reviewer #1 #2, #3, and #4 have carefully revised our manuscript accordingly.

We have conducted a study to analyze foodborne pathogens, particularly parasites, that may arise due to climate change and dietary habits involving the consumption of raw foods. In this study, we aimed to identify optimal genetic testing methods for detecting parasites that may cause acute or chronic infections in foods of concern in the future, based on a review and comparison of domestic and international literature on real-time genetic analysis methods. We believe that the real-time PCR method can provide faster results than conventional PCR methods, which is why we chose this approach for our research. Additionally, the PCR composition was based on methods previously used by the Korea Food and Drug Administration for genetic testing of parasites during foodborne illness investigations.

Regarding the target of this paper, Anisakis simplex, there have been reports of it in food in Korea in 2017, 2019, and 2023. Therefore, as the reviewers mentioned, it would be very challenging to analyze actual samples from food. Consequently, we determined that it would be appropriate to use samples obtained from another research institution in Korea, the Whale Research Institute, which included five samples of Anisakis simplex and two samples of Pseudoterranova decipiens.

---

## [Decision Letter · Decision Letter 1]

3 Dec 2024

PONE-D-24-33370R1Comparative study of Taqman-based real-time PCR assay for the detection of Anisakis simplex and Pseudoterranova decipiensPLOS ONE

Dear Dr. Seo,

Thank you for submitting your manuscript to PLOS ONE. After careful consideration, we feel that it has merit but does not fully meet PLOS ONE’s publication criteria as it currently stands. Therefore, we invite you to submit a revised version of the manuscript that addresses the points raised during the review process.

The authors should improve the material and methods part,  such as primer design, sample collection and qPCR validation, with improved respective discussion of results and conclusion.

Please submit your revised manuscript by Jan 17 2025 11:59PM. If you will need more time than this to complete your revisions, please reply to this message or contact the journal office at plosone@plos.org . Please include the following items when submitting your revised manuscript:

We look forward to receiving your revised manuscript.

Kind regards,

Elingarami Sauli, PhD

Academic Editor

PLOS ONE

Journal Requirements:

Reviewers' comments:

Reviewer's Responses to Questions

**Comments to the Author**

1. If the authors have adequately addressed your comments raised in a previous round of review and you feel that this manuscript is now acceptable for publication, you may indicate that here to bypass the “Comments to the Author” section, enter your conflict of interest statement in the “Confidential to Editor” section, and submit your "Accept" recommendation.

Reviewer #2: All comments have been addressed

Reviewer #3: (No Response)

Reviewer #5: (No Response)

Reviewer #6: (No Response)

2. Is the manuscript technically sound, and do the data support the conclusions?

Reviewer #2: Yes

Reviewer #3: Partly

Reviewer #5: Partly

Reviewer #6: Partly

3. Has the statistical analysis been performed appropriately and rigorously? 

Reviewer #2: Yes

Reviewer #3: Yes

Reviewer #5: No

Reviewer #6: No

4. Have the authors made all data underlying the findings in their manuscript fully available?

Reviewer #2: Yes

Reviewer #3: Yes

Reviewer #5: No

Reviewer #6: No

5. Is the manuscript presented in an intelligible fashion and written in standard English?

Reviewer #2: Yes

Reviewer #3: Yes

Reviewer #5: No

Reviewer #6: No

6. Review Comments to the Author

Reviewer #2: Thank you for taking my comments into account in the manuscript.

Please also change line 185 in the Results and discussion chapter to the correct accession number of the genomic sequences mtDNA (a. simplex KC810002) (P. decipiens OK338708)

Reviewer #3: Manuscript Number: PONE-D-24-33370R1

Comparative study of Taqman-based real-time PCR assay for the detection of Anisakis simplex and Pseudoterranova decipiens

Comments: -

The manuscript has shown improvement since the first revision. However, it still requires significant enhancements to meet publication standards. Below are my comments and suggestions for further refinement:

Results and Discussion: While the results section is improved, the manuscript would benefit from a more in-depth discussion. Expanding the discussion section will enhance the manuscript's robustness and make it more suitable for publication. Consider addressing broader implications, limitations, and potential future research directions.

qPCR Section: It is essential for the authors to adhere to the Minimum Information for Publication of Quantitative Real-Time PCR Experiments (MIQE) guidelines (Bustin et al., 2009). Please ensure compliance with these guidelines and include a completed MIQE checklist in the supplementary data to strengthen the methodological transparency and reliability.

Line 45: It is recommended that the nomenclature adheres to the MIQE guidelines. Specifically, the term "quantification cycle (Cq)" should be used for consistency and clarity.

Line 130: The term "Universal mater mix" should be corrected to "2X Universal Master Mix."

Line 133: The term "PCR amplification" should be revised to "qPCR amplification" as the instrument used is a real-time PCR system.

Lines 134 & 135: The term "PCR condition" would be more appropriately replaced with "thermocycle condition."

Line 137: The nomenclature for "real-time PCR" should adhere to the MIQE guidelines. Specifically, use "qPCR" for quantitative real-time PCR and "RT-qPCR" for reverse transcription-qPCR. Please ensure this correction is applied consistently throughout the manuscript.

Line 175: The phrase "thirteen detection methods" is confusing and could be misleading. It would be more appropriate to use "thirteen sets of primers/probes" for clarity and accuracy.

Lines 181-183: The statement "According to Figure 6..." requires clarification regarding the algorithm used in the analysis. Was it Maximum Likelihood (ML), Neighbor-Joining (NJ), or Bayesian inference? Please specify the algorithm in the statement for better transparency and understanding.

Reviewer #5: The authors have addressed a useful tool, but the manuscript is not well drafted.

Major concerns

1. The authors should thoroughly review methods for nematode detection in introduction part, such as reverse PCR, LAMP, ddPCR. There must be successful methods in model nematode species.

2. The materials and methods part should be rephrased, such as primer design, sample collection and qPCR validation. The positive control is a part of sample collection. The authors should list all sample used in the manuscript, such as how and where the samples were collected. The authors should collect marine samples affected with nematode to validate the method. Inter-examinator validation belongs to technological repeats, which should be added to the part of qPCR. Besides, a melt curve should be carried out for each pair of primers.

3. Figures and tables are cited confusedly, which should be arranged in turn.

4. The results part should be rephrased according to the materials and methods part in new revision.

5. The conclusion part is too long. Some parts should be moved to discussion part.

Reviewer #6: The manuscript “Comparative study of Taqman-based real-time PCR assay for the detection of Anisakis simplex and Pseudoterranova decipiens” describes Taqman qPCR-based assay for the detection of two parasites. There are several points that the authors need to address:

Language Issues:

1- The authors need to revise the language of the manuscript. There are instances of wrong word order and sentence structure.

a. Line 60-61: … infected raw fish infected

b. Line 67: comma is not needed here-… symptoms, such as…

c. Line 73: do not use semi-colon. Use full-stop - …reported; thus…

Methods:

Line 119-121: it is not clear which probe concentration is for which target.

Line 147: it seems a 5 has been missed here (10-5 -fold (1ⅹLOD) and 10-5 -fold 148 (5ⅹLOD))

In general:

Since the authors prepared standards, they need to provide Limit of Quantification (LOQ). The LOS provided does not seem correct. The researchers need to first determine the LOQ and then determine LOD, based on a 2-fold serial dilution.

In the materials and methods section, no sensitivity study has been mentions. The authors need to point out the sensitivity study.

Did the author try real samples? Did they use approve positive and negative samples?

The author conducted inter-examiner study. However, they need to conduct inert-assay and intra-assay experiments.

7. PLOS authors have the option to publish the peer review history of their article (what does this mean? ). If published, this will include your full peer review and any attached files.

**Do you want your identity to be public for this peer review?** For information about this choice, including consent withdrawal, please see our Privacy Policy .

Reviewer #2: No

Reviewer #3: No

Reviewer #5: No

Reviewer #6: No

---

## [Author Response · Author response to Decision Letter 1]

13 Jan 2025

I would like to express my gratitude for the time both reviewers and editor have dedicated to review my paper. I have endeavored to incorporate comments as much as possible, and have provided comprehensive revisions in Response to Reviewers.

---

## [Decision Letter · Decision Letter 2]

25 Feb 2025

Comparative study of Taqman-based qPCR assay for the detection of Anisakis simplex and Pseudoterranova decipiens

PONE-D-24-33370R2

Dear Dr. Doo Win Sen,

We’re pleased to inform you that your manuscript has been judged scientifically suitable for publication and will be formally accepted for publication once it meets all outstanding technical requirements.

Kind regards,

Elingarami Sauli, PhD

Academic Editor

PLOS ONE

Additional Editor Comments (optional):

The authors have satisfactorily addressed all reviewer comments to allow acceptance of this submission, but this should be upon addressing/adhering to the journal guidelines/format, including addressing grammatical issues.

Reviewers' comments:

Reviewer's Responses to Questions

**Comments to the Author**

1. If the authors have adequately addressed your comments raised in a previous round of review and you feel that this manuscript is now acceptable for publication, you may indicate that here to bypass the “Comments to the Author” section, enter your conflict of interest statement in the “Confidential to Editor” section, and submit your "Accept" recommendation.

Reviewer #5: All comments have been addressed

2. Is the manuscript technically sound, and do the data support the conclusions?

Reviewer #5: Yes

3. Has the statistical analysis been performed appropriately and rigorously? 

Reviewer #5: Yes

4. Have the authors made all data underlying the findings in their manuscript fully available?

Reviewer #5: Yes

5. Is the manuscript presented in an intelligible fashion and written in standard English?

Reviewer #5: Yes

6. Review Comments to the Author

Reviewer #5: The authors have addressed most of the comments. The manuscript could be accepted after grammar checking.

7. PLOS authors have the option to publish the peer review history of their article (what does this mean? ). If published, this will include your full peer review and any attached files.

**Do you want your identity to be public for this peer review?** For information about this choice, including consent withdrawal, please see our Privacy Policy .

Reviewer #5: No

---

## [Editor Report · Acceptance letter]

PONE-D-24-33370R2

PLOS ONE

Dear Dr. Seo,

I'm pleased to inform you that your manuscript has been deemed suitable for publication in PLOS ONE. Congratulations! Your manuscript is now being handed over to our production team.

Kind regards,

on behalf of

Dr. Elingarami Sauli

Academic Editor

PLOS ONE